# Synthesis, Characterization, Biological Evaluation and DNA Interaction Studies of 4-Aminophenol Derivatives: Theoretical and Experimental Approach

**DOI:** 10.3390/molecules27041352

**Published:** 2022-02-17

**Authors:** Bushra Rafique, Saima Kalsoom, Abdulrahim A. Sajini, Hammad Ismail, Mudassir Iqbal

**Affiliations:** 1Department of Chemistry, School of Natural Sciences, National University of Sciences and Technology (NUST), Islamabad 44000, Pakistan; bushra.rafique@sns.nust.edu.pk; 2Department of Chemistry, Preston University, Islamabad 44000, Pakistan; genious_chemist@yahoo.com; 3Healthcare Engineering Innovation Center (HEIC), Department of Biomedical Engineering, Khalifa University, Abu Dhabi P.O. Box 127788, United Arab Emirates; abdulrahim.sajini@ku.ac.ae; 4Department of Biochemistry and Biotechnology, Umar Al Khayyam Block, Hafiz Hayat Campus, University of Gujrat, Gujrat 50700, Pakistan; hammad.ismail@uog.edu.pk

**Keywords:** Schiff base derivatives, DNA interaction, antimicrobial studies, antidiabetic studies, molecular docking

## Abstract

In the present study, five 4-aminophenol derivatives (4-chloro-2-(((4-hydroxyphenyl)imino)methyl)phenol(S-1), 4-((4-(dimethylamino)benzylidene)amino)phenol(S-2), 4-((3-nitrobenzylidene)amino)phenol(S-3), 4-((thiophen-2-ylmethylene)amino)phenol(S-4) and 4-(((*E*)-3-phenylallylidene)amino)phenol(S-5)) were synthesized and characterized by FT-IR, ^1^H-NMR, ^13^C-NMR and elemental analyses. The synthesized compounds were tested for their antimicrobial (Gram-positive and Gram-negative bacteria and *Saccharomyces* *cervesea* fungus) and antidiabetic (α-amylase and α-glucosidase inhibitory) activities. All the compounds showed broad-spectrum activities against the *Staphylococcus aureus* (ATCC 6538), *Micrococcus luteus* (ATCC 4698), *Staphylococcus epidermidis* (ATCC 12228), *Bacillus subtilis sub. sp spizizenii* (ATCC 6633), *Bordetella bronchiseptica* (ATCC 4617) and *Saccharomyces cerevisiae* (ATCC 9763) strains. The newly synthesized compounds showed a significant inhibition of amylase (93.2%) and glucosidase (73.7%) in a concentration-dependent manner. Interaction studies of Human DNA with the synthesized Schiff bases were also performed. The spectral bands of S-1, S-2, S-3 and S-5 all showed hyperchromism, whereas the spectral band of S-4 showed a hypochromic effect. Moreover, the spectral bands of the S-2, S-3 and S-4 compounds were also found to exhibit a bathochromic shift (red shift). The present studies delineate broad-spectrum antimicrobial and antidiabetic activities of the synthesized compounds. Additionally, DNA interaction studies highlight the potential of synthetic compounds as anticancer agents. The DNA interaction studies, as well as the antidiabetic activities articulated by the molecular docking methods, showed the promising aspects of synthetic compounds.

## 1. Introduction

Schiff bases are condensation products of primary amines with carbonyl compounds. The azomethine group (RHC=N–R_1_) is a common structural feature of these compounds (where R and R_1_ = alkyl, aryl, cyclo alkyl or heterocyclic groups). The other names of such compounds are anils, imines or azomethines. Various studies [1,2,3] showed that the presence of a lone pair of electrons in a sp^2^ hybridized orbital of a nitrogen atom (N) of the azomethine group is, basically, of considerable importance. The relative ease of preparation, synthetic flexibility and the unique properties of the C=N group make Schiff bases excellent chelating agents [4], especially in the cases where functional groups, such as –OH or –SH, are present close to the azomethine group that can facilitate the formation a five- or six-membered ring with the metal ion. The versatility of Schiff base ligands and the analytical, biological and industrial applications of their complexes make further investigations in this area highly desirable.

Schiff bases have various applications in different fields of life. Salicylaldehyde-based Schiff bases have been reported as plant growth regulators with antimicrobial or antimycotic activity. Schiff bases with hydroxyl-substituents have been known to exhibit good pharmaceutical [5,6] and antimicrobial properties [7]. Schiff bases also show some analytical applications [8]. One of the interesting applications of Schiff bases is their use as corrosion inhibitors. This property is, basically, their ability to spontaneously form a monolayer on the surface to be protected. Most of the commercial inhibitors available are aldehyde- or amine-based, but the C=N bond of the Schiff bases plays this role more effectively in many cases [9]. The principal interaction between the inhibitor and the metal surface is chemisorption [10]. The inhibitor molecule should have centers capable of forming bonds with the metal surface by electron transfer. In such cases, the metal acts as an electrophile and the inhibitor acts as a Lewis base. The nucleophilic centers, such as oxygen and nitrogen atoms, of the protective compound have free electron pairs, which are readily available for sharing. Together with the atoms of the benzene rings, they create multiple absorption sites for the inhibitor, thus enabling stable monolayer formation [11]. The –N=CH– (imine) group of Schiff bases is mainly involved in elucidating the mechanism of transamination and the racemization reaction in biological systems [12]. An imine linkage between the aldehyde derived from vitamin A and the protein opsin in the retina of the eye plays an important role in the chemistry of vision. Vitamin B6 serves as a coenzyme by forming an imine with an amino acid grouping with an enzyme. The coenzyme, bound to the enzyme, is involved in the transamination reaction, the transfer of the amino group from one amino acid to another, which is important in the metabolism and biosynthesis of amino acids. In the last step, enzyme-catalyzed hydrolysis cleaves the imine into a pyridoxal and the modified amino acid [8]. Schiff bases are active against a wide range of micro-organisms [13,14]. 4-dimethylamine benzaldehyde Schiff bases showed antibacterial activity [15]. Metformin is used as antidiabetic medication. Metformin (antidiabetic drug) has benefits in other diseases, including cancer, infectious diseases, aging-related diseases etc. The potential effect of metformin as an adjuvant therapy to antibiotics was also investigated recently. Metformin exhibited an antibacterial effect when combined with the antibiotics on the tested strains and exhibited low toxicity on the mammalian cells [16]. This paper presents a series of Schiff bases with a potential biological activity resulting from the acid-catalyzed condensation of aldehydes with aromatic amines. These compounds could also act as valuable ligands. Such kinds of Schiff bases have more biological potential because of the free electron delocalization associated with the ring structure [17]. Schiff bases derived from heterocyclic rings present many advantages, and the most predominant heteroatoms found in organic molecules are mainly nitrogen (N), oxygen (O), and sulfur (S) [18]. As it is genetic material and contains all cellular information, the study of DNA interaction is a very important aspect at present. The interaction of DNA with small molecules and their different binding modes are already reported [19,20,21]. The interactions between small molecules and DNA are of two types, covalent interactions and non-covalent interactions. The outcomes of our current studies showed that the synthesized Schiff base compounds were having various non-covalent types of interactions with DNA. There are three major types of non-covalent binding interactions, i.e., electrostatic, groove and intercalative. Electrostatic binding occurs as a result of the negatively charged phosphate backbone of the DNA interaction with the positively charged ends of small molecules. Major and minor groove binding are the two modes of groove binding. In groove binding, hydrogen bonding or van der Waals interactions of the small molecule with the nucleic acid bases can occur. Intercalation arises by the insertion of molecules within the nucleic acid–base pairs [22,23]. Various biologically important small molecules interact with DNA through non-covalent binding interactions [24,25]. These small molecules are known to have specific binding modes with DNA. However, the literature showed mixed or more than one binding mode for different small molecules with DNA. This property of binding mode can be linked to their mechanism of action and therapeutic efficiency [26]. Such kinds of studies have attracted more attention for the designing of more effective drugs which can target DNA.

The integrated management of multiple diseases simultaneously can be achieved by using single compound rather than multiple drugs. Therefore, reducing toxicity is important for exploring the new areas in drug discovery. In a continuation of the above ideas, the related data was recently published where multiple antibiotics, along with an antidiabetic compound, were tested on mammalian cells [16]. The results showed that antidiabetic compounds can act synergistically with antimicrobial potential and reduced toxicity. Inspired by the above facts, a series of 4-aminophenol Schiff’s bases was synthesized in the current study and characterized by elemental analysis and spectral (IR, ^1^H- and ^13^C-NMR) analyses. The synthesized Schiff base derivatives were also evaluated for their potential biological activities, i.e., antibacterial, antidiabetic and DNA interaction studies, etc. 

## 2. Materials and Methods

### 2.1. General

5-chlorosalisaldehyde, 4-*N*,*N*-dimethylaminobenzaldehyde, 2-thiophencarboxyaldehyde, 3-nitrobenzaldehyde, cinnamaldehyde, acarbose, potassium Iodide (KI), iodine (I_2_), potassium dihydrogen phosphate, dipotassium hydrogen phosphate were obtained from Sigma Aldrich (Hamburg, Germany), *p*-nitrophenyl β-d-glucopyranoside (*p*-NPG), α-glucosidase, methanol from Merck (KGaA, Darmstadt, Germany), starch, NaHCO_3_, hydrochloric acid (HCl), glacial acetic acid, absolute ethanol and dimethylsulfoxide (DMSO) from Sigma Aldrich (St. Louis, MO, USA).

### 2.2. Instrumentation and Measurements

Melting points were determined in open capillaries on a FALC Instrument SRL360D. TLC (Silica Gel 60 F254) from Merck (Beijing, China) was used to check the compounds’ purities. Elemental analyses were carried out by using Euro EA 3000 (EUROVECTOR, Pavia, Italy) instrument. The infrared spectra of the Schiff bases were recorded on FT-IR (Bruker Optics GmbH & Co. Ettlingen, Germany) with an ATR model (ALPHA 200488) in the frequency range of 4000–550 cm^–1^. NMR spectra were recorded in DMSO-*d*_6_ and CDCl_3_ solvent on a BRUKER AV-500 MHz NMR (Bruker Beijing Scientific Technology, Beijing, China). A high-resolution multinuclear FT-NMR spectrometer was used in the presence of SiMe_4_ (internal standard) at δ = 0 ppm at 25 °C.

The sbsorbance of α-amylase and α-glucosidase activities (at 540 and 405 nm, respectively) were measured on a microplate reader (BioTek Elx-800, Agilent Technologies, Santa Clara, CA, USA). An InnuPREP Blood DNA Mini Kit (Analytik Jena) (AJ Innuscreen GmbH, Berlin, Germany) was used for the extraction of genomic DNA from human blood. A PDA spectrophotometer (Agilent 8453) Agilent Technologies, Berkshire, UK, along with a quartz optical cell (with an effective absorption path of 1 cm), was used for DNA interaction studies of the synthesized Schiff bases.

### 2.3. General Synthetic Procedure of Schiff Bases

A series of five Schiff bases of 4-aminophenol (4-chloro-2-(((4-hydroxyphenyl)imino)methyl)phenol (S-1), 4-((4-(dimethylamino)benzylidene)amino)phenol (S-2), 4-((3-nitrobenzylidene)amino)phenol (S-3), 4-((thiophen-2-ylmethylene)amino)phenol (S-4) and 4-(((*E*)-3-phenylallylidene)amino)phenol (S-5) were synthesized by adopting the following general procedure. To the best of our knowledge, the Schiff base S-1 was synthesized for the first time in our work. This general procedure was adapted, with slight changes for the S-2 to S-5 compounds [27,28,29,30]. Both reagents were individually mixed in absolute ethanol, and reaction mixtures were stirred at room temperature (r.t) (10–30 min) where glacial acetic acid (2–3 drops) was used as a catalyst. The monitoring of reaction progress was performed by TLC. The obtained precipitates were dried after filtration. Absolute ethanol was used for recrystallization and the resulting crystals/powder were dried in a vacuum. The obtained products were weighed for their actual yields. The reaction conditions are mentioned in Figure 1 for all the synthesized 4-aminophenol Schiff base derivatives.

### 2.4. Antimicrobial Activity

The agar-well diffusion method was applied to evaluate the antimicrobial properties of the synthesized compounds and their comparison with control compounds by following the National Committee for Clinical Laboratory Standards (M07-A8).

Different bacterial and fungal strains were investigated to determine the in vitro antimicrobial activities of the compounds. American type culture was selected, and the *Staphylococcus aureus* (ATCC 6538), *Micrococcus luteus* (ATCC 4698), *Staphylococcus epidermidis* (ATCC 12228), *Bacibacillus subtilis sub. sp spizizenii* (ATCC 6633), *Bordetella bronchiseptica* (ATCC 4617) and *Saccharomyces cerevisiae* (ATCC 9763) strains were used in the present study. Optimal growth conditions were adopted for the culture growth of microbes in a sterilized nutrient agar medium. Surface cleanliness was performed by using a 0.9% NaCl solution for inoculum preparation. No activities were observed for the negative control (ethanol). Stock solutions (1 mg/mL) of the synthesized compounds were prepared in ethanol. This is the minimum inhibitory concentration (MIC), as preliminary tests were performed on different low concentrations (0.5 mg/mL, 0.75 mg/mL and 1 mg/mL) for all the Schiff bases. The MIC was observed to be 1 mg/ ml. Metronidazole was used as a positive control for the antibacterial activity, and Nystatin was used for the antifungal activity comparison. Previously prepared inoculums of microbial cultures were mixed with the freshly prepared media. Antimicrobial activity was investigated by inoculating a 100 µL solution of each compound solution in the respective agar well plates. The plates were allowed 24 h (at 37 °C) for incubation. An evaluation of the antimicrobial activities was conducted using a ProtoCOL3 zone reader by measuring the inhibition zone in millimeters (mm).

### 2.5. Antidiabetic Studies

In vitro antidiabetic studies were performed using α-amylase and α-glucosidase inhibition assays.

#### 2.5.1. α-amylase Inhibition Assay

The previously reported procedure was adopted to perform the assay [31]. Transparent 96-well microtiter plates were used in the procedure. The sequential order of components to perform the α-amylase inhibition activity was as follows: 40 μL of 50 mM potassium phosphate buffer (pH 6.8), 10 μL of the sample at different concentrations, 10 μL of α-amylase (0.12 U/mL) and 40 μL of 0.05% starch. Acarbose served as a positive control, and DMSO served as a negative control. The resulting reaction mixture was incubated for 30 min at 50 °C. Then 20 μL of 1M HCl was poured to stop the reaction. In order to investigate either in the presence or absence of starch, 80 μL of iodine reagent (5 mM KI and 5 mM I_2_) was added at the end of procedure. The percentage of inhibition and the IC_50_ values were calculated using GraphPad Prism V8 (GraphPad Software, LLC, San Diego, CA, USA).

#### 2.5.2. α-glucosidase Inhibition Assay

A previously reported method with minor modifications was adopted to perform the α-glucosidase inhibitory activity [31]. The sequential order to add the chemicals in the reaction mixture prepared in a 96-well plate was as follows: 25 μL of *p*-nitrophenyl β-d-glucopyranoside (*p*-NPG) (20 mM), 69 μL of phosphate buffer (50 mM, pH 6.8), 1 μL of α-glucosidase enzyme (3 U/mL) and 5 µL of sample, with final concentrations of 800, 400, 200, 100, 50, 25, 12.5 and 6.25 μg/mL, were added in respective wells. Acarbose served as a positive control, and DMSO served as a negative control. The incubation was carried out for 30 min at 37 °C temperature. Then, the reaction was stopped by adding 100 µL of NaHCO_3_ (0.5 mM) and the absorbance was recorded using a microplate reader (BioTek Elx-800, Agilent Technologies, Santa Clara, CA, USA) at 405 nm. The percentage of inhibition and the IC_50_ values were calculated using GraphPad Prism V8 (GraphPad Software, LLC, CA, USA).

### 2.6. DNA Interactions Studies

The interaction with DNA of the test compound was studied by using the UV spectrophotometer method reported in literature with small modifications [32]. In this assay, an innuPREP Blood DNA Mini Kit (Analytik Jena) (AJ Innuscreen GmbH, Berlin, Germany) was used for the extraction of genomic DNA from human blood. The assay was conducted on a PDA spectrophotometer (Agilent 8453) Agilent Technologies, Inc., UK. Methanol and water, in a ratio of 9:1, were used as a blank and solvent to dissolve the compounds. The spectrum of the pure compound was recorded, and, to this, DNA was added in different concentrations, step wise, starting from lowest concentration to higher concentrations, keeping the compound concentration in the reaction mixture constant. The absorbance measurements were recorded by a UV-visible spectrophotometer by keeping the concentrations of compounds constant (100 µM) in the sample cell while varying the concentrations of DNA (0.5 × 10^−6^ to 1 × 10^−6^, 0.5 × 10^−5^ and 1 × 10^−5^ M) in the sample cell. The spectra were recorded in the form of spectral peaks in each step and the change in the absorbance was noted. To achieve an equilibrium between the compound and the DNA, the solutions were allowed to stay for at least 5 min before each measurement was made. 

### 2.7. Molecular Docking Studies

The 3D structures of the DNA (PDB id 1BNA) [33] and the antidiabetic target 2ZE0 [34] were obtained from the Protein Data Bank (www.rcsb.org/pdb, accessed on 7 July 2021) and then targeted atoms were prepared, charged, protonated and minimized via MOE 2016 suite Chemical Computing Group, Köln, Germany [35]. The 3D structures of the synthesized Schiff bases were built and saved in their 3D conformations by the Builder tool incorporated in MOE 2016. Further protonation, minimization, charge application and atom-type corrections were also performed. The docking protocol was validated by re-docking studies. The different parameters for the docking studies were used and fixed, such as energy minimization, gradient: 0.005, force field: MMFF94X+Solvation and chiral constraint: current geometry. Energy minimization was terminated when the root mean square gradient fell below 0.05. The minimized structure of the DNA was used as the template for docking. Ten conformations were generated for each docked compound DNA and ligand 2ZE0 target complex.

## 3. Results and Discussion

### 3.1. Analysis of Physical Data

The synthesized compounds were of different colors. The physical data of the 4-aminophenol Schiff bases are presented in Table 1.

### 3.2. Microanalysis

The elemental analysis data of the synthesized Schiff bases is presented in Table 2. The observed results were found in agreement with the theoretical calculations. The thin-layer chromatography (TLC) and elemental (C, H and N) analyses were used to check the purity of the compounds.

### 3.3. Spectroscopic Studies

#### 3.3.1. FT-IR (Fourier Transform Infrared) Spectroscopy

In the IR spectrum of 4-aminophenol (used as the amine for the synthesis of all the Schiff bases with different aldehydes), the phenolic ν(O–H) stretching band was observed at 3338 cm^−1^, and N–H stretching was observed at 3282 cm^−1^. The IR spectrum of the ligand exhibited an absorption band at 1613 cm^−1^, which can be assigned to ν(N–H) bending in the amine. Ν(O–H) bending was observed at 1384 cm^−1^, and ν(C–N) stretching was observed at 1235 cm^−1^. ν(C–O) stretching was observed at 1091 cm^−1^ for 4-aminophenol (Table 3). The Schiff base formation was confirmed by comparing the IR spectra of the Schiff base ligands and their respective reagents. 

An important doublet in the C–H stretching region for aldehyde C–H ~ 2916 and ~ 2854 cm^−1^ was missed in the synthesized Schiff base (S-1). The Schiff base exhibited an IR band at 1656 cm^−1^ that was attributed to the C=N stretching frequency (imine group) as compared to the C=O stretching frequency in free aldehyde at 1663 cm^−1^. The peaks at 1618 and 1564 were attributed to the ν(C=C) stretching bands in the Schiff base S-1.

The 1659 cm^−1^ band was assigned to ν(C=O) stretching in 4-*N*,*N*-dimethylaminobenzaldehyde and was shifted to 1608 cm^−1^ (C=N) imine formation after the condensation reaction. The band at 1581 cm^−1^ was assigned to the stretching vibrations of ν(C=C) in the synthesized Schiff base (S-2). The methyl groups attached to the substituted nitrogen atom at position 4 showed a stretching vibration of ν(C–N) at 1160 cm^−1^ in the reacting aldehyde. 

The 1701 cm^−1^ band that was assigned to ν(C=O) stretching in 3-nitrobenzaldehyde was shifted to 1654 cm^−1^ for (C=N) azomethine group formation in the synthesized compound (S-3). The aromatic nitro compound showed asymmetric stretching for ν(N=O) at 1530 cm^−1^ and symmetric stretching at 1348 cm^−1^ for the aldehyde. These bands were present at the same value in the synthesized Schiff base, showing no participation of a nitro group in bond formation.

The 1654 cm^−1^ band that was assigned to ν(C=O) stretching in 2-Thiophencarboxyaldehyde was shifted to 1608 cm^−1^ for ν(C=N) azomethine group formation in the synthesized compound (S-4). For the sharp band at 863, ν(C–S–C) of a thiophene moiety in aldehyde remained almost the same in the synthesized S-4 Schiff base.

The band at 1668 cm^−1^ that was assigned to ν(C=O) stretching in cinnamaldehyde was shifted to 1643 cm^−1^ for ν(C=N) azomethine group formation in the synthesized S-5 Schiff base. The 1628 cm^−1^ band corresponded to the stretching vibrations of ν(C=C) in aldehyde.

#### 3.3.2. Nuclear Magnetic Resonance (NMR) Spectroscopy

Proton (^1^H) and Carbon (^13^C)-NMR

The ^1^H-NMR spectra of the Schiff bases were determined and were found to be consistent with the expected structures (Table 4). A singlet around 9.72–9.31 ppm was assigned to the –OH proton of aminophenol. A sharp singlet at around 8.89–8.38 ppm was assigned to the azomethine proton (C=N) in the spectra of all the Schiff bases. Around 8.66–6.74 ppm, observed multiplets were assigned to the aromatic protons. The 3.05 ppm spectrum was assigned to the protons present in the form of N(CH_3_)_2_ in the S-2 Schiff base. 

The sharp singlet around 162.26–152.24 ppm was assigned to the azomethine carbons (–C=N) in all the Schiff bases in the ^13^C-NMR spectra. The aromatic carbon signals mostly appeared in the range of 152.49–111.98 ppm for aromatic compounds. The spectrum at 130.90 ppm was assigned to the C–Cl present in the S-1 compound, the spectrum at 40.16 ppm was assigned to the –CH_3_ carbon of S-2, the spectrum at 142.11 ppm was assigned to the carbon attached to the nitro group (–NO_2_) in the S-3 compound, the spectra at 143.04 and 127.83 ppm were assigned to the carbons attached to the hetero atom (–S) present in the ring of the S-4 compound and the spectra at 134.61 and 129.40 ppm were assigned to the aliphatic carbons of the S-5 compound.

### 3.4. Antimicrobial Studies

A remarkable increase in human pathogenic bacteria was seen in recent decades, due to resistance to a number of antimicrobial agents [37]. The exploration of new antimicrobial drugs with better resistance has become the main objective for scientists [38]. The resistance of bacteria to the synthesized Schiff bases can be associated with:The enzymatic degradation of the synthesized compound;An alteration of the bacterial protein targeted by the synthesized compound;A change in the membrane permeability to the tested compound, etc. [39].

Additionally, Gram-negative bacteria are found to be very troublesome and have modified themselves to nullify the effects of almost all antibiotics and antimicrobial choices available [40]. These molecules form the bilayer membrane to control and regulate the flux of molecules in the cells. The complex lipids present in Gram-negative bacteria may stop the diffusion of chemicals into the cytoplasm of organisms. This may not be the case in Gram-positive bacteria. Therefore, this ability makes them more resistant to chemicals compared to Gram-positive bacteria. 

Interestingly all the synthesized compounds showed moderate to strong activities against both Gram-positive and Gram-negative bacteria [41,42]. A comparison of the antimicrobial activities of the 4-aminophenol Schiff bases with antibacterial (Metronidazole) and antifungal (Nystatin) control drugs is shown in Table 5. S-1 showed better results against *S. aureus* in our study (14.18 mm inhibition zone at a concentration of 1 mg/mL in ethanol) as compared to that reported in the literature (where it showed 30 mm inhibition zone at a concentration of 20 mg/mL in DMSO solvent) [43]. S-2 showed the best results, comparatively, against *S.aureus* and *M.luteus* and least activity was shown against these bacteria by Schiff base S-1. 

Similarly, S-4 showed moderate activity against *S*. *epidermidis,* and S-3 was found to have strong activity against *B. spizizenii.* Meanwhile, the substitution on the phenyl ring with polar groups (Nitro) led to lower values of the lipophilicity parameters [44]. The shift of the nitro group in the compound S-3 led to a reduction in the antibacterial effects against most of tested microorganisms. However, S-4 also showed strong activity against *B. bronchiseptica*. Moreover, all the synthesized compounds also showed good activity against the tested fungus, i.e., *Saccharomyces cerevisiae*. The present study showed that the antifungal activity of the S-4 Schiff bases was almost equivalent to the reference drug. However, S-5 was found to be more active as compared to the reference drug Nystatin. Cinnamaldehyde is a major constituent of cinnamon (essential oils) and has potent antifungal activities against a wide variety of fungi [45,46,47]. The obtained results can be inspired by researchers for the synthesis of new potential bioactive compounds. The graphical comparison of the synthesized Schiff bases for their antimicrobial activities is shown in Figure 1.

### 3.5. Antidiabetic Studies

The synthesized compounds were evaluated in the α-amylase inhibition assay to detect activity, which might be extrapolated to detect a potential antidiabetic effect using a different concentration. The percentage of inhibition displayed by each compound is shown in Table 6. The a-amylase activity was observed in S-2, S-1, S-3 and S-5 under in vitro conditions at 500 ppm. However, the S-4 activity against α -amylase was negligible. Among these compounds, S-2 showed comparatively better activity than the other tested compounds at this concentration. Additionally, the α-glucosidase inhibitory activity was also observed to further confirm the antidiabetic activities of these tested compounds (Table 7). The compound S-1 showed the maximum percentage of inhibitory activity (76.67%) among all compounds at concentration of 500 ppm. Likewise, in terms of IC_50_, S-2 showed the lowest values, representing a strong anti-amylase potential as compared with the other compounds. Overall, the results exhibited that these compounds possess moderate antidiabetic potential in a concentration-dependent manner.

### 3.6. DNA Interaction Studies

A single compound may utilize more than one mode of DNA binding (e.g., intercalation and groove binding) [48]. Hence, in addition to exploring the bioactivity of our prepared Schiff bases, we were also interested to study their DNA binding mechanisms. Different experimental techniques can be utilized to study the drug–DNA binding interactions and distinguish the binding modes (intercalation from groove binding). UV-Vis spectroscopy, viscometry, fluorescence spectroscopy, etc., are among the principal techniques used.

UV–visible spectroscopy serves as one of the basic techniques to study small molecule–DNA interactions. Two major features of the spectra of DNA are the ‘‘hyperchromic’’ effect and the ‘‘hypochromic effect’’ that arise due to a change in its double helical structure [49]. The absorbance spectra of the compounds showed hyperchromism (increased absorbance intensity) upon titration with varying amounts of DNA, suggestive of compound interaction with DNA (Figure 2). Hyperchromism arises from the breaking down of the DNA secondary structure [50]. In the present study, the spectral bands of the S-1, S-2, S-3 and S-5 compounds showed hyperchromism. However, the spectral bands of the S-2, S-3 and S-4 compounds also showed a bathochromic shift (red shift). The first response (the lowest absorbance in the case of S-1, S-2, S-3 and S-5 and the highest absorbance in case of S-4) in the spectrum of each interaction is purely of the ligand or Schiff base (100 µM in each case). When a molecule bound with DNA and helix stabilization increases by the insertion of flat aromatic species between base pairs a hypochromic effect can be observed. A hypochromic effect was observed in the case of the S-3 compound’s interaction with DNA.

The conformational and structural changes that occur in DNA on binding with small molecules can be translated into a change in its spectral behavior. A destabilization of the secondary structure of DNA occurs when DNA interacts with small molecules, which leads to hyperchromism, whereas hypochromism instigates from the stabilization of the DNA secondary structure via an electrostatic or intercalation binding mode of small molecules [51,52]. Generally, the intercalation of small molecules into the DNA double helix shows a bathochromic shift (red shift), as well as a hypochromic effect. It is important to mention that the absorption spectra of DNA may be influenced by drugs or vice versa. To remove ambiguity, a baseline correction with a suitable concentration of a drug or DNA must be performed before studying the interaction using UV-visible spectroscopy 

In DNA non-covalent interactions with small molecules, intercalation and minor groove binding are the most common interactions [53]. Intercalation implies the stacking insertion of a planar molecule between the layers of stacked bases in a double-stranded DNA. While it does not directly damage DNA, the DNA-intercalator complex inhibits the activity of topoisomerase enzymes involved in DNA replication processes [54]. Intercalation reduces DNA’s helical twist and lengthens the DNA [55].

Ethidium Bromide (EB) is a well-known intercalator that binds tightly to DNA and is used to tag DNA, due to its strong fluorescent nature in various biological experiments [56]. In the intercalation phenomenon, DNA conformation alterations occur.

The *λ*_max_ for S-1 was found to be at a wavelength of 357 nm. In the current study, no considerable *λ*_max_ shift was observed after the interaction of the compound with different concentrations of DNA. However, the hyperchromic effect was observed with increasing concentrations of DNA. This type of interaction is considered to be intercalative interaction.

The *λ*_max_ for the S-2 compound was observed at 202 nm. A considerable *λ*_max_ shift towards a higher wavelength was observed in this case. This shift in *λ*_max_ was due to the DNA adduct formation with the S-2 compound. With an increase in the DNA concentration, a red shift (bathochromic shift) can be observed. Moreover, as in the case of the S-1 complex with DNA, a hyperchromic shift was also observed here with increasing concentrations of DNA. Thus, the compound S-2 showed intercalative or strong interactions with DNA [57]. In the case of S-3, the *λ*_max_ was observed at 201 nm, and the compound, again, showed hyperchromism and a bathochromic shift (201–213 nm) with intercalative interaction. However, in case of S-4, the *λ*_max_ was observed at 341 nm. Here, a slight wavelength shift (341–346 nm) or a small bathochromic shift was observed, but, interestingly, the hypochromic effect (decrease in intensity) was also observed with an increase in the concentration. The compactness in the structure of either the compound and/or the DNA after complex formation (between the synthesized compound and DNA) may result in hypochromism [58]. In case of the intercalation mode of DNA binding with a compound, the evident spectral feature of the band under study is hypochromism, with a small shift in the wavelength; occasionally, no shift is observed. This is because the intercalative type of interaction comprises a significant interaction between the aromatic chromophore of the complex compounds and the DNA base pairs [59]. S-5 showed a *λ*_max_ at 351 nm, and no prominent shift in wavelength was observed. However, the hyperchromic effect was, again, observed with intercalative attraction mode between the Schiff base and the DNA. These observations showed the possibilities of changes in the conformation and structure of DNA after the interaction has occurred. The hyperchromic effect is actually the increase in absorbance of the compound/DNA after interaction. Stacking interactions held together two strands of DNA via hydrogen bonding or a hydrophobic effect between the complementary bases. The hydrogen bond limits the resonance of the aromatic ring, so the absorbance of the sample is limited as well. Furthermore, the hyperchromic effect may also be ascribed to external contact via electrostatic binding [60] or via a partial uncoiling of the DNA helix structure, thus exposing more DNA bases [61]. If only weak interactions exist in the compound interactions with DNA, only hypochromic or hyperchromic effects can be observed and no significant changes of peak shifts in the spectral data can be observed [62,63]. Thus, four out of the five tested compounds showed hyperchromism at all concentrations, whereas one compound (S-4) showed the hypochromism effect (Table 8).

The hyperchromic effect might be attributed to the external contact (electrostatic binding) [64] or partial uncoiling of the helix structure of DNA, which resulted in the exposure of more DNA bases [31]. The hyperchromic effect may also be due to the electrostatic interaction between positively charged moieties in the interacting compound and the negatively charged phosphate backbone present at the periphery of the DNA double-helix [32].

### 3.7. Molecular Modeling Behavior

Antidiabetic docking studies were also run for these synthesized compounds. All synthesized compounds showed strong binding interactions (H-binding and Arene-π) interactions with the target except the S-4 compound. The S-4 docked view showed the binding mode of this compound at a lower site of proximity contour with no binding interactions, as shown in Figure 3. The S-2 compounds showed both hydrophilic and hydrophobic interactions (Figure 4) with the target and may be responsible for its high alpha amylase activity. Similarly, S-1, S-3 and S-4 also showed strong hydrogen binding with key contributing residues. Their docked binding energy values ranged from −12.321 to −10.51 kcal/mol. Therefore, we assume that this hydrogen bonding interaction and binding patterns might be responsible for the binding affinities of these Schiff bases against the diabetic target. 

Drug–DNA binding studies were performed by using a docking protocol. All of the five compounds intercalated in the double helix DNA strand, as shown in Figure 5. Strong H-binding was observed between the electronegative center of the ligands and the DC A11, DC A9 and DT B20, nitrogen bases. These docking poses strongly support the experimental findings of the Schiff base DNA results, as shown in Table 8. 

## 4. Conclusions

Five Schiff base derivatives (S-1 to S-5) were synthesized in the present study. The physical properties and microanalysis of the synthesized compounds were evaluated, and spectral studies were performed using FT-IR and NMR techniques. The biological screening was performed by antimicrobial, antidiabetic and DNA interaction studies. S-2 showed the best results against *S*. *aureus* and *M. luteus*. S-3 was found to have strong activity against *B. spizizenii.* The S-4 Schiff base also showed strong activity against *B*. *bronchiseptica*. Moreover, all the synthesized compounds also showed good activity against the tested fungus, i.e., *S*. *cerevisiae*. S-5 was found to be more active as compared to the reference drug, Nystatin. Conclusively, all the tested compounds showed broad-spectrum activities against the tested microorganisms. The antidiabetic activities of the synthesized Schiff bases were also evaluated, and the results showed that the S-1 to S-5 compounds have α-amylase inhibitory activities under in vitro conditions, ranging from 93.19–36.14% at the concentration of 500 ppm. Among them, compound S-2 showed a comparatively better α-amylase inhibitory activity than the other tested compounds at this concentration. The S-1 compound showed the maximum percentage of α-glucosidase inhibitory activity (76.67%) among all the compounds at the maximum applied concentration (500 ppm). DNA interaction studies of 4-aminophenol Schiff base derivatives were also performed against human blood DNA. This interaction study showed that the spectral bands of the S-1, S-2, S-3 and S-5 compounds showed hyperchromism, whereas the spectrum of only the S-4 compound showed hypochromism. However, the spectral bands of the S-2, S-3 and S-4 compounds also exhibited a bathochromic shift (red shift) when they formed complexes with DNA after interactions. The results indicate that the reaction of the 4-aminophenol compound with different aldehydes yielded Schiff base derivatives with promising antibacterial, antifungal and DNA binding activities. A good correlation was found between the molecular docking, experimental antidiabetic and DNA results and, therefore, may prove useful for further lead optimization and clinical trials. However, these compounds could not show good antidiabetic activities, as the results showed. In the future, these synthesized compounds can be recommended for medicinal purposes after comprehensive investigations of the determined properties.

## Data Availability

Data can be made available upon request from authors.

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
