# Peer review of "Synthesis, Characterization, Biological Evaluation and DNA Interaction Studies of 4-Aminophenol Derivatives: Theoretical and Experimental Approach"

_molecules, 2022, doi:10.3390/molecules27041352_

Round 1

Reviewer 1 Report

I still feel that the paper is not suitable for publication despite all the comments were addressed.  I just feel it is beyond the level of the Journal as the experiments are too simlistic and results are too superficial, approximate and have a little further use.

I do not like the approach to the study of interactions between ligand and DNA, though it is quite popular nowadays. It cannot be fixed however; only the new experiment can help.

The accuracy of measurements of inhibition zones seems fantastic as well as claime errors - they are actually tens of micrometers! Despite the experimental results, I do not believe that the studied compounds have any therapevtic potential.

However, the work had been done, the experiments had been performed, the report had been written, the questions had been addressed. I have no formal reason to reject the paper.

Reviewer 2 Report

The paper can be accepted in present form.

This manuscript is a resubmission of an earlier submission. The following is a list of the peer review reports and author responses from that submission.

Round 1

Reviewer 1 Report

The paper "Synthesis, Chracterization, Biological Evaluation and DNA In-2 teraction Studies of 4-Aminophenol Derivatives: Theoretical 3 and Experimental Approach" is devoted to the synthesis, spectral characterisation, primary anti-microbial activity and DNA-binding testing of few Schiff bases.

The main problem of the proposed paper is the lack of novelty. Despite Author's claim "Schiff bases S-1 was synthesized for the first 131 time in our work.", all the studied compounds are long known (some of them even from 19th century!) [Journal of the American Pharmaceutical Association (1912), 1955, vol. 44, p. 148], [Journal of the American Chemical Society, 1998, vol. 120, # 28, p. 7107 - 7108], [Dalton Transactions, 2011, vol. 40, # 2, p. 421 - 430], [Journal of Heterocyclic Chemistry, 2011, vol. 48, # 5, p. 1067 - 1072], [Revue Roumaine de Chimie, 2005, vol. 50, # 7-8, p. 641 - 648], [Chemisches Zentralblatt, 1907, vol. 78, # I, p. 108], [Journal of the Indian Chemical Society, 2009, vol. 86, # 12, p. 1338 - 1342], [Journal of the Chemical Society, 1908, vol. 93, p. 537], [Acta Crystallographica, Section C: Crystal Structure Communications, 2000, vol. 56, # 8, p. 1044 - 1045], [Journal of Organic Chemistry, 1949, vol. 14, p. 790,795], [Chemische Berichte, 1892, vol. 25, p. 2753][Chemische Berichte, 1893, vol. 26, p. 394], [Medicinal Chemistry Research, 2018, vol. 27, # 3, p. 807 - 816].

The antimicrobial activity of the compounds is hard to estimate without standard deviations of the inhibition zone diameters; moreover, the control experiment with pure ethanol is missing. The antidiabetic studies show that acarbose is way more effective than any of studied compounds, so why bother? 

"The absorbance spectra of compounds showed hyper-344 chromism (increased absorbance intensity) upon titration with varying 345 amounts of DNA, suggestive of compound interaction with DNA."
Disagree. To prove that some interaction occurs, it is necessary to show that the spectrum of DNA+ligand mixture is not equal of the spectra of DNA and ligand registered separately. Why on Fig. 2 absorbance drops below zero in all cases in the long wavelenth interval? The compounds emits light instead of absorbing? The quality of Figures is low and requires improvement. The study of DNA binding is too superficial.

Molecular docking is too unreliable and rarely predicts modes of molecules correctly [Int J Mol Sci. 2016 Apr; 17(4): 525], [J. Chem. Inf. Model. 2021, 61, 6, 2957–2966].

I recommend to reject this paper as it contains too little new information; and that, which is new, is superficial and unreliable.
